# Dietary Nutritional Supplements Are Associated with the Deterioration of Hepatic Fibrosis in Women and Individuals without Underlying Disease

**DOI:** 10.3390/ijerph21101298

**Published:** 2024-09-28

**Authors:** Minsu Cha, Sangheun Lee, Kijun Han

**Affiliations:** 1Department of Emergency Medicine, International St. Mary’s Hospital, Catholic Kwandong University College of Medicine, Incheon 22711, Republic of Korea; 605214@ish.ac.kr; 2Institute of Health and Medical Convergence Research, International St. Mary’s Hospital, Catholic Kwandong University College of Medicine, Incheon 22711, Republic of Korea; 3Department of Internal Medicine, International St. Mary’s Hospital, Catholic Kwandong University College of Medicine, Incheon 22711, Republic of Korea; 545818@ish.ac.kr; 4Division of Hepatology, International St. Mary’s Hospital, Catholic Kwandong University College of Medicine, Incheon 22711, Republic of Korea

**Keywords:** dietary nutritional supplements, hepatic fibrosis, FIB-4, APRI

## Abstract

Despite the growing societal interest in the health benefits of dietary nutritional supplements, their safety and efficacy remain unclear. We aimed to investigate the correlation between hepatic fibrosis and the consumption of dietary nutritional supplements. This study utilized data from the Korea National Health and Nutrition Examination Survey spanning the period from 2014 to 2022. Significant fibrosis was defined as a fibrosis index based on four factors (FIB-4) ≥1.45 and an aspartate aminotransferase-to-platelet ratio index (APRI) ≥0.30. Adjusted odds ratios (AORs) and 95% confidence intervals (CIs) were calculated. In a study involving 30,639 participants (supplement consumers [*n* = 17,772] and non-consumers [*n* = 12,867]), dietary nutritional supplement consumption was associated with alanine aminotransferase (ALT) elevation and increased hepatic fibrosis biomarkers (APRI and FIB-4). Dietary nutritional supplement consumption was independently linked to ALT elevation (AOR, 1.11; 95% CI, 1.04–1.18), FIB-4 (AOR, 1.07; 95% CI, 1.00–1.15), and APRI (AOR, 1.14; 95% CI, 1.07–1.21). This association was particularly significant in women and subgroups of people who were not diabetic or hypertriglyceridemic. In our comprehensive analysis, the consumption of dietary nutritional supplements was possibly associated with hepatic fibrosis, particularly in specific subgroups. Given the limitations of this study, these findings are not considered definitive conclusions; however, they serve as valuable preliminary data for future research.

## 1. Introduction

Dietary nutritional supplements can be defined and classified as a broad spectrum of nutraceuticals, including vitamins, minerals, dietary elements, food components, natural herbs, herbal preparations, and synthetic compounds [1]. They can generally be obtained without a prescription and taken without specific medical advice or monitoring. Due to the limited clinical trial requirements for dietary nutritional supplements compared to conventional drugs, the safety and efficacy of these supplements have not been thoroughly evaluated, leading to potential safety concerns [2].

According to a study examining trends in overall and micronutrient-containing dietary nutritional supplement use among adults and children in the United States, the prevalence of dietary nutritional supplement use increased from 50% in 2007 to 56% in 2018 [3]. Interestingly, the use of dietary nutritional supplements has increased over the last 20 years, with a notable surge in sales globally, particularly during the coronavirus disease 2019 (COVID-19) pandemic. The rise in supplement intake, especially during the pandemic, was primarily driven by the expectation that these supplements could aid in boosting immunity, thereby preventing COVID-19 infection [4]. However, research on how and to what extent dietary nutritional supplements enhance immunity, as well as evidence supporting their role in preventing COVID-19, remains insufficient. Despite this, the interest in and sales of dietary nutritional supplements increased significantly during this period.

A study published in the United States indicated that dietary nutritional supplements are sold over the counter without sufficient verification and that such supplements may sometimes interfere with other prescribed medications, occasionally leading to serious side effects. This study suggests that regulations and oversight are necessary to address these issues [2]. For example, Stickel et al. critically reviewed the literature on liver injury associated with dietary nutritional supplementation. They found that significant liver injury was associated with the intake of certain dietary nutritional supplements [5]. Although it cannot be generalized that all dietary nutritional supplements adversely affect liver health, the examples provided by Stickel et al. highlight the potential risks associated with dietary nutritional supplements that can be purchased without specific regulations or medical consultation.

Hepatic fibrosis is a major health concern in South Korea. A recent study indicated that the sex- and age-standardized prevalence of significant and advanced hepatic fibrosis amongst metabolic dysfunction-associated fatty liver disease was 9.7% and 3.0%, respectively [6]. Recurring liver damage from various causes, including hepatitis B and C and fatty liver disease, can lead to hepatic fibrosis, which involves different cellular pathways, predominantly hepatic stellate cells [7,8,9]. Left untreated, hepatic fibrosis can evolve into cirrhosis, which is characterized by severe structural damage to the liver, impaired liver function, and the development of complications such as portal hypertension, liver failure, and hepatocellular carcinoma. While the liver has a remarkable capacity to regenerate and repair damage, advanced fibrosis severely compromises its regenerative ability and significantly reduces life expectancy [10]. However, there is an ongoing debate regarding the efficacy of dietary nutritional supplements among well-nourished individuals, particularly concerning their impact on chronic diseases, including hepatic fibrosis. This study investigated the correlation between hepatic fibrosis and dietary nutritional supplements sold in South Korea using data obtained from the Korea National Health and Nutrition Examination Survey (KNHANES).

## 2. Materials and Methods

### 2.1. Participant Eligibility

KNHANES is a cross-sectional survey conducted by the Korea Centers for Disease Control and Prevention to assess health-related behaviors, chronic disease prevalence, dietary intake, and nutritional status in a representative Korean population. We used data from KNHANES VI–IΧ spanning the period from 2014 to 2022. Ethical review and approval were waived as the data used are publicly available. Informed consent was obtained from all survey participants prior to participation, and all survey protocols received approval from the National Center for Health Statistics Ethics Review Board.

### 2.2. Participants

A total of 68,023 participants were registered in KNHANES between 2014 and 2022, and a health survey reported the intake of dietary nutritional supplements in the participants over the past year. We excluded participants who were aged <19 years (n = 12,700), those who tested positive for the hepatitis B surface antigen (n = 6131) and the hepatitis C antibody (n = 417), and significant alcohol consumers (n = 3293). Participants with missing socioeconomic information, including residential district, educational level, and household income, were excluded (n = 2249). Participants with missing lifestyle information, including smoking status, alcohol consumption, and aerobic activity, were also excluded (n = 328). Participants with missing essential clinical information, such as dietary nutritional supplement intake (n = 5427), body mass index (BMI) (n = 235), and diabetes mellitus (n = 899), hypertension (n = 138), and hypertriglyceridemia (n = 5451) status, were excluded from the analysis. Participants with missing essential blood test data, including alanine aminotransferase (ALT), aspartate aminotransferase (AST), and platelet counts, were excluded (n = 116). Ultimately, 30,639 participants were included in the study (Figure 1).

### 2.3. Measurements of Clinical and Biochemical Variables

Smoking status was classified as “never smoked”, “ex-smoker”, or “current smoker.” Heavy alcohol consumption was defined as >210 g ethanol/week for men and >140 g ethanol/week for women. Aerobic activity was defined as engaging in >150 min of moderate-intensity physical activity or >75 min of vigorous-intensity physical activity per week. The BMI was calculated by dividing the weight (kg) by the height squared (kg/m^2^). The average of two to three blood pressure measurements was recorded. Hypertension was defined as systolic blood pressure >140 mmHg or diastolic blood pressure >90 mmHg or the use of antihypertensive medication. Venous blood samples were collected for testing after fasting for 8 h, and fasting blood sugar, AST, ALT, triglyceride, and cholesterol levels were measured. Individuals with fasting blood sugar levels >126 mg/dL or those who had previously been diagnosed with diabetes by a doctor were classified as having diabetes. Hypertriglyceridemia was defined as a triglyceride level ≥200 mg/dL or the individual being on medications for hypertriglyceridemia.

### 2.4. Definition of Dietary Nutritional Supplements and Hepatic Fibrosis

Dietary nutritional supplements include tablets, capsules, powders, granules, and liquids that contain vitamins, minerals, and functional ingredients. Participants who had consumed dietary supplements for over 2 weeks in the past year were defined as dietary nutritional supplement consumers. Herbal medicines and decoctions prescribed by doctors in Korean medicine were excluded.

Serologic biomarkers, such as the fibrosis index based on four factors (FIB-4) and the AST-to-platelet Ratio Index (APRI), were employed to determine the presence of hepatic fibrosis [11,12]. FIB-4 was calculated as follows: FIB-4 = (age [years] × AST [IU/L])/(platelet count [10^9^/L] × √ALT [IU/L]). An FIB-4 value ≥1.45 indicated significant hepatic fibrosis [10]. The APRI was calculated as follows: (AST/AST upper limit of normal/platelet count [10^9^/L]) × 100. An APRI value ≥0.3 was considered indicative of significant hepatic fibrosis [13]. Moreover, significant ALT elevation was defined as ALT ≥33 IU/L for men and ALT ≥25 IU/L for women [14].

### 2.5. Statistical Analysis

Descriptive statistics were used to outline the baseline characteristics of the study population. Independent t-tests and chi-squared tests were used to compare continuous and categorical variables, respectively. To evaluate the association between hepatic fibrosis and the use of dietary nutritional supplements, we conducted a multivariate logistic regression analysis, with a *p*-value < 0.05 considered statistically significant. In our analysis, we adjusted for age (continuous), sex, residential district, educational level, household income, BMI (continuous), smoking status (never smoked, ex-smoker, and current smoker), aerobic activity, and comorbidities (diabetes mellitus, hypertension, hypertriglyceridemia, and hypercholesterolemia). The study population was stratified into two groups depending on sex, presence of diabetes mellitus, or hypertriglyceridemia to control for the effects of sex or underlying metabolic disease. R software (version 4.4.1; R Foundation for Statistical Computing, Vienna, Austria) was used for all statistical analyses.

## 3. Results

### 3.1. Participants’ Baseline Characteristics

Of the total participants, 12,419 (40.5%) were men, and 18,220 (59.4%) were women. Among the men, 6405 (51.5%) reported consuming dietary nutritional supplements. The men who consumed nutritional supplements were slightly older (53 years) and included a higher proportion of urban residents (80.8%), college-educated individuals (46.6%), and those in higher-income brackets (38.5%) compared to the men in the control group. The prevalence of hypertension was higher in supplement consumers compared to the control group; however, the rates of diabetes mellitus and hypertriglyceridemia were similar between the groups. Notably, supplement users had a lower proportion of current smokers (25.1%) and a higher prevalence of BMI ≥ 23 kg/m^2^ (67.6%). Table 1 summarizes the demographic and clinical characteristics of the men in this study.

Among the women, 11,367 (62.3%) reported consuming dietary nutritional supplements. These women tended to be older, with a mean age of 53 years, compared to the control group. A greater proportion of supplement users lived in urban areas (82.6%) compared to those in the control group (79.3%). Additionally, more supplement consumers had a college education (37.1%) and belonged to higher-income brackets (29.9% in the high-income category). While the prevalence of hypertension and hypertriglyceridemia was comparable between both groups, the rates of diabetes mellitus were lower among supplement consumers (11.1%). Supplement users had a slightly lower percentage of individuals with a BMI ≥ 23 kg/m^2^ (50.0%). Table 2 provides detailed demographic and clinical characteristics of the women in this study.

### 3.2. Association between Dietary Nutritional Supplement Consumption and ALT, FIB-4, and APRI

The assessment of hepatic fibrosis using the FIB-4 index revealed that a higher proportion of dietary nutritional supplement consumers (27.5%) had scores indicative of fibrosis (≥1.45) than the participants in the control group (26.2%), with a statistically significant difference (*p* < 0.01). Furthermore, the APRI score, another marker for hepatic fibrosis, was higher in the dietary nutritional supplement group (20.4%) than in the control group (19.0%), again with a significant difference (*p* < 0.01). These findings suggest that dietary nutritional supplement consumption is associated with higher incidences of hepatic fibrosis, indicating a potential area for closer scrutiny of the long-term effects of dietary nutritional supplements on liver health.

In our data, dietary nutritional supplement consumption was associated with elevated ALT levels, with thresholds of ALT ≥33 for men and ALT ≥25 for women. The adjusted odds ratio (AOR) for dietary nutritional supplement consumption was 1.11 (95% confidence interval [CI]: 1.04–1.18, *p* < 0.01). Using FIB-4 as the reference, dietary nutritional supplement consumption was independently associated with FIB-4 ≥ 1.45, with an AOR of 1.07 (95% CI: 1.00–1.15, *p* < 0.05). Similarly, dietary nutritional supplement consumption was independently associated with an APRI score ≥ 0.30, with an AOR of 1.14 (95% CI: 1.07–1.21, *p* < 0.001).

### 3.3. Association between Hepatic Fibrosis and Dietary Nutritional Supplement Consumption Stratified by Diabetes Mellitus, Hypertriglyceridemia, and Sex

To investigate the relationship between dietary nutritional supplement intake and hepatic fibrosis, we stratified the cohort based on the presence of diabetes mellitus, hypertriglyceridemia, and sex. Using the APRI and FIB-4 scores as fibrosis markers, with APRI ≥ 0.3 and FIB-4 ≥ 1.45 indicating significant fibrosis, we observed the following: In the subgroup without diabetes mellitus, the prevalence of significant fibrosis, as indicated by an APRI score ≥ 0.3 and FIB-4 ≥ 1.45, was consistently higher in participants who consumed dietary nutritional supplements than in participants in the control group, with notable differences (18.7% vs. 16.7% of APRI ≥ 0.3 [*p* < 0.001] and 25.1% vs. 22.9% of FIB-4 ≥ 1.45 [*p* < 0.001]). Dietary nutritional supplement consumption was independently associated with ALT elevation (AOR, 1.10; 95% CI, 1.02–1.19), FIB-4 ≥1.45 (AOR, 1.11; 95% CI, 1.02–1.20), and APRI ≥0.30 (AOR, 1.17; 95% CI, 1.09–1.26) in participants without diabetes mellitus (Figure 2).

This trend was also observed in participants without hypertriglyceridemia (19.6% vs. 17.4% of APRI ≥ 0.3 [*p* < 0.001] and 28.3% vs. 26.7% of FIB-4 ≥ 1.45 [*p* < 0.01]). Dietary nutritional supplement consumption was independently associated with ALT elevation (AOR, 1.11; 95% CI, 1.03–1.20), FIB-4 ≥ 1.45 (AOR, 1.10; 95% CI, 1.02–1.19), and APRI ≥ 0.30 (AOR, 1.19; 95% CI, 1.11–1.27) in participants without hypertriglyceridemia (Figure 3). In participants with diabetes mellitus or hypertriglyceridemia, hepatic fibrosis (APRI ≥ 0.3 or FIB-4 ≥ 1.45) was not independently associated with dietary nutritional supplement consumption in the multivariate logistic regression analysis.

The cohort was stratified by gender. Women who consumed dietary nutritional supplements exhibited a consistently higher prevalence of significant fibrosis than those in the control group, with 16.5% vs. 13.6% for APRI ≥ 0.3 (*p* < 0.001) and 26.5% vs. 24.28% for FIB-4 ≥ 1.45 (*p* < 0.001). The multivariate logistic regression analysis for women indicated that dietary nutritional supplement consumption was independently associated with ALT elevation (AOR, 1.13; 95% CI, 1.03–1.24), FIB-4 ≥ 1.45 (AOR, 1.12; 95% CI, 1.02–1.23), and APRI ≥ 0.30 (AOR, 1.18; 95% CI, 1.08–1.29) (Figure 4). The prevalence of significant fibrosis was higher in men who consumed dietary nutritional supplements than in those in the control group for APRI ≥ 0.3 (27.4% vs. 25.3%, *p* < 0.01). However, for FIB-4 ≥ 1.45, the difference was not statistically significant (29.5% vs. 28.4%, *p* = 0.202). The independent association between nutritional supplement consumption and hepatic fibrosis was not significant in men.

## 4. Discussion

Our study revealed a significant association between liver health and the consumption of dietary nutritional supplements. In particular, participants who consumed dietary nutritional supplements had significantly higher levels of APRI and FIB-4, both indicators of hepatic fibrosis, than those in the control group. Hepatic fibrosis is a major pathological condition characterized by the gradual build-up of extracellular matrix proteins, leading to liver scarring. Fibrosis progression can result in cirrhosis and increase the risk of hepatocellular carcinoma. Although treatments for hepatitis B and C can reverse the fibrosis caused by these viruses, there are no direct treatments available for fibrosis induced by toxins or metabolites. Therefore, providing dietary nutritional supplements with medical guidance is crucial, as they can produce harmful toxins or metabolites.

Non-invasive methods to assess hepatic fibrosis, such as elastography, have advanced significantly and offer alternatives to liver biopsies. Recent advances have introduced artificial intelligence to diagnose liver fibrosis [15]. If these methods are not feasible, the two most commonly used non-invasive scoring systems are FIB-4 and APRI. In our study, the FIB-4 score was calculated using age, AST and ALT levels, and platelet counts, whereas the APRI score combined the AST levels and the platelet counts. Both scoring systems serve as alternatives to transient elastography, with specific cut-off values indicating significant fibrosis or cirrhosis. Although further validation is needed to confirm their diagnostic accuracy, these tools are especially useful in settings where carrying out liver measurements is not feasible.

Dietary nutritional supplement consumption was independently associated with elevated ALT levels and an increased risk of hepatic fibrosis (APRI ≥ 0.3 and FIB-4 ≥ 1.45), indicating a need for careful consideration of the long-term effects of dietary nutritional supplements on liver health. Nevertheless, these results should be interpreted with caution because some of the 95% CIs were close to 1, suggesting a potentially weak association. The results of the multivariate analyses indicated that the 95% CIs of the subgroups were less close to 1 than those of the overall population. Therefore, instead of advising caution regarding dietary nutritional supplement consumption for the general population, focusing on specific subgroups for which this association is stronger may be more pertinent.

This study found that, among the participants without diabetes mellitus or hypertriglyceridemia, dietary nutritional supplement users had a higher risk of significant fibrosis (APRI ≥ 0.3 and FIB-4 ≥ 1.45) compared to non-users. These findings suggest that dietary nutritional supplement consumption is associated with an increased risk of hepatic fibrosis, especially in individuals without diabetes mellitus or hypertriglyceridemia. A potential cause for the lack of association between dietary nutritional supplement consumption and hepatic fibrosis in patients with diabetes or hypertriglyceridemia could be the overwhelming fibrogenic impact of cardiometabolic risk factors. Cardiometabolic risk factors, such as diabetes mellitus and hypertriglyceridemia, are strongly associated with hepatic fibrosis and may overshadow any statistical association between dietary nutritional supplements and hepatic fibrosis. Considering the widespread use of dietary nutritional supplements, these results emphasize the need for cautious use, particularly in individuals without diabetes mellitus or hypertriglyceridemia. The long-term consumption of dietary nutritional supplements without specific reasons may sometimes lead to unintended hepatic fibrosis.

Dietary nutritional supplement consumption was independently associated with a higher risk of hepatic fibrosis (APRI ≥ 0.3 and FIB-4 ≥ 1.45) in women, but not in men. The observed differences in the effects of nutritional supplements on hepatic fibrosis between men and women could be attributed to various sex-specific factors. Sexual dimorphism in adipose tissue distribution and substrate metabolism significantly affects insulin sensitivity and cardiometabolic health, both of which are strongly associated with hepatic steatosis and fibrosis [16,17,18]. Hormonal influences may also have contributed to these results [19]. Furthermore, a comprehensive analysis of 54 studies concluded that women have a lower risk of developing nonalcoholic fatty liver disease (NAFLD) but a higher risk of advanced fibrosis once NAFLD is established compared to men [20].

It is important to note that not all research indicates that dietary nutritional supplements worsen hepatic fibrosis. A recent experimental study has demonstrated that certain combination of metabolic cofactors can improve liver conditions such as NAFLD and hepatic fibrosis in mice [21]. Additionally, dietary fiber therapy has been shown to alleviate liver fibrosis, and a botanical supplement derived from white peony and licorice was found to attenuate NAFLD by modulating the gut microbiota and reducing inflammation in mice [22,23]. Although these studies highlight the potential for dietary nutritional supplements to mitigate hepatic fibrosis, these findings are based on experimental studies using animal models, and it remains uncertain whether they are both effective and safe in humans. In recent studies involving human participants, curcumin, naringenin, and propolis have shown protective effects against hepatic fibrosis and steatosis in NAFLD patients [24,25,26]. The studies involving human participants were randomized, placebo-controlled, prospective clinical trials that yielded promising results for the development of future therapies. However, the trials involved only modest numbers of participants and were conducted specifically on NAFLD patients, leaving uncertainty about whether these supplements would have similar liver health benefits for healthy individuals.

This study had several limitations. First, as this was a cross-sectional study, it could not establish causality between dietary nutritional supplement use and hepatic fibrosis. Additionally, selection bias may have occurred. For example, the users of dietary nutritional supplements tended to be older, more educated, reside in urban areas, and were predominantly women compared to non-users. However, KNHANES is representative of national survey data, with a large sample size. The high correlation between dietary nutritional supplements and hepatic fibrosis is a significant finding. Moreover, we attempted to overcome these limitations by performing logistic regression analysis, adjusting for various factors, such as age, sex, educational level, residential district, household income, smoking status, BMI, aerobic activity, and comorbidities, to elucidate the relationship between dietary nutritional supplements and hepatic fibrosis. Second, this study did not specifically classify the types and contents of dietary nutritional supplements, which may have varied among participants. Further research to determine the specific dietary nutritional supplements associated with hepatic fibrosis is warranted, as it will provide comprehensive insights into the potential risks posed by different types of supplements. Third, careful interpretation of the results is warranted because some of the 95% CIs were close to 1, suggesting a weak association. Finally, our study did not include liver biopsy or elastography results, which are considered the gold standard and widely used alternative techniques for diagnosing hepatic fibrosis, respectively. However, due to the high costs and potential complications associated with biopsies, non-invasive diagnostic methods such as the FIB-4 and APRI scores have been widely adopted and accepted in recent research.

## 5. Conclusions

The findings of our extensive data analysis suggest a potential association between dietary nutritional supplement consumption and the development of hepatic fibrosis, particularly in women and individuals without diabetes mellitus or hypertriglyceridemia. However, due to this study’s limitations, including its retrospective nature and the lack of detailed information on supplement contents, these results are preliminary and not conclusive. Further research is required to understand these findings, inform guidelines for the safe consumption of dietary nutritional supplements, and clarify the mechanisms underlying this association, including the role of dosage and the duration of use.

## Figures and Tables

**Figure 1 ijerph-21-01298-f001:**
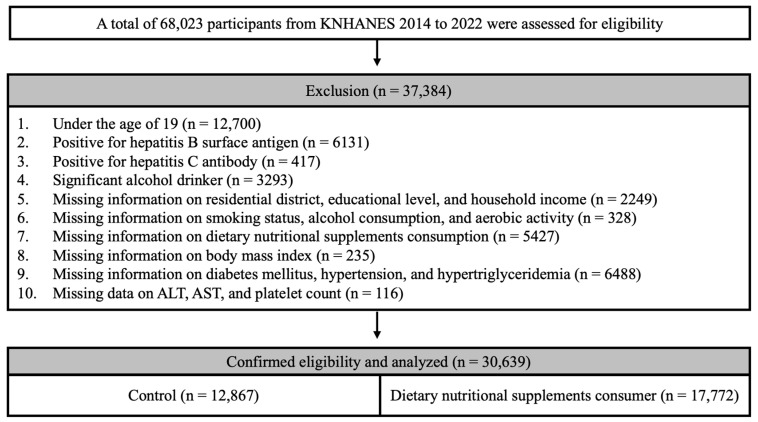
Flowchart of participant selection from KNHANES 2014 to 2022.

**Figure 2 ijerph-21-01298-f002:**
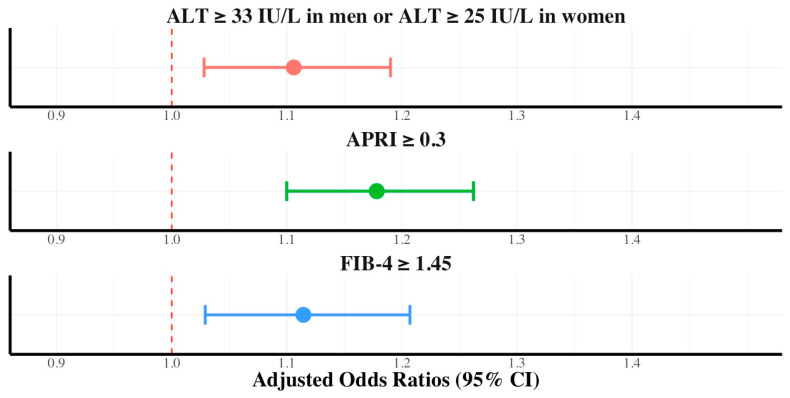
Adjusted odds ratios with 95% confidence intervals of dietary supplements in ALT, FIB-4, and APRI for participants without diabetes mellitus.

**Figure 3 ijerph-21-01298-f003:**
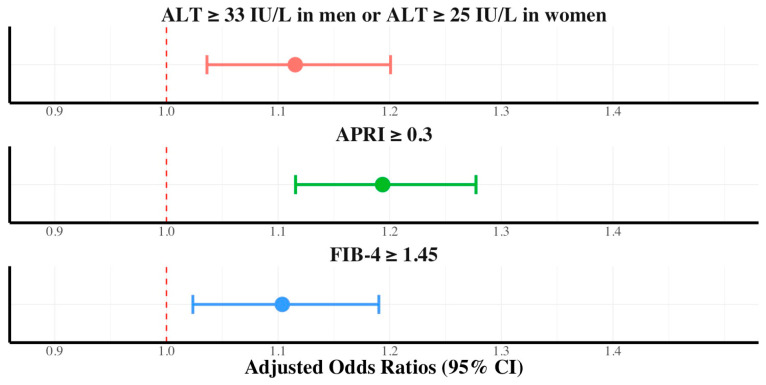
Adjusted odds ratios with 95% confidence intervals of dietary supplements in ALT, FIB-4, and APRI for participants without hypertriglyceridemia.

**Figure 4 ijerph-21-01298-f004:**
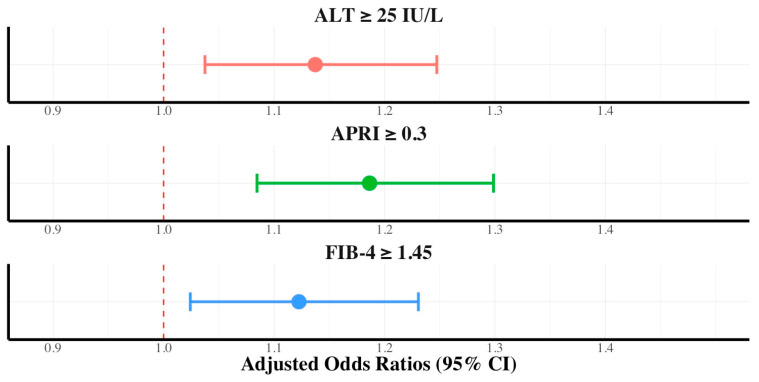
Adjusted odds ratios with 95% confidence intervals of dietary supplements in ALT, FIB-4, and APRI in women.

**Table 1 ijerph-21-01298-t001:** Demographic and clinical characteristics of the men participants.

Variables	Control (n = 6014)	Dietary Nutritional Supplement Consumer (n = 6405)	*p*-Value
Age (years)	51 ± 18	53 ± 16	<0.001
Residential district			<0.001
Urban	4705 (78.2)	5179 (80.8)	
Rural	1309 (21.7)	1226 (19.1)	
Education level			<0.001
Elementary school	1084 (18.0)	766 (11.9)	
Middle school	651 (10.8)	568 (8.8)	
High school	2188 (36.3)	2084 (32.5)	
College	2091 (34.7)	2987 (46.6)	
Household income			<0.001
Low	1290 (21.4)	977 (15.2)	
Mid-low	1555 (25.8)	1462 (22.8)	
Mid-high	1665 (27.6)	1768 (27.6)	
High	1504 (25.0)	2198 (34.3)	
Hypertension	2190 (36.4)	2467 (38.5)	<0.05
Diabetes mellitus	983 (16.3)	1017 (15.8)	0.494
Hypertriglyceridemia	1119 (18.6)	1217 (19.0)	0.590
Smoking			<0.001
Never smoked	1602 (26.6)	1658 (25.8)	
Ex-smoker	2437 (40.5)	3138 (48.9)	
Current smoker	1975 (32.8)	1609 (25.1)	
BMI (kg/m^2^) ≥ 23	3966 (65.9)	4336 (67.6)	<0.05
Waist circumference (cm)	86.9 ± 9.7	87.4 ± 8.8	<0.05
Aerobic activity rate	2777 (46.1)	3230 (50.4)	<0.001
Fasting glucose (mg/dL)	103.4 ± 24.9	103.2 ± 22.8	0.780
Triglycerides (mg/dL)	147.6 ± 114.6	148.5 ± 107.0	0.672
HDL-C (mg/dL)	47.2 ± 11.4	47.9 ± 11.3	<0.001
LDL-C (mg/dL)	114.5 ± 35.5	116.3 ± 34.6	0.087
AST (IU/L)	24.6 ± 13.7	25.2 ± 12.0	<0.05
ALT (IU/L)	26.3 ± 20.7	27.1 ± 20.0	<0.05
FIB-4 ≥ 1.45	1713 (28.4)	1892 (29.5)	0.202
APRI ≥ 0.30	1524 (25.3)	1756 (27.4)	<0.01

Continuous variables are expressed as means and standard deviations. Categorical variables are expressed as numbers and percentages. Abbreviations: BMI, body mass index; HDL, high-density lipoprotein cholesterol; LDL, low-density lipoprotein cholesterol; ALT, alanine aminotransferase; AST, aspartate aminotransferase; FIB-4, fibrosis index based on four factors; and APRI, AST-to-platelet ratio index.

**Table 2 ijerph-21-01298-t002:** Demographic and clinical characteristics of the women in the study.

Variables	Control (n = 6853)	Dietary Nutritional Supplement Consumer (n = 11,367)	*p*-Value
Age (years)	51 ± 18	53 ± 15	<0.001
Residential district			<0.001
Urban	5437 (79.3)	9393 (82.6)	
Rural	1416 (20.6)	1974 (17.3)	
Education level			<0.001
Elementary school	2054 (29.9)	2463 (21.6)	
Middle school	600 (8.7)	1163 (10.2)	
High school	2006 (29.2)	3523 (30.9)	
College	2193 (32.0)	4218 (37.1)	
Household income			<0.001
Low	1655 (24.1)	2076 (18.2)	
Mid-low	1764 (25.7)	2714 (23.8)	
Mid-high	1768 (25.7)	3168 (27.8)	
High	1666 (24.3)	3409 (29.9)	
Hypertension	2152 (31.4)	3554 (31.2)	0.860
Diabetes mellitus	888 (12.9)	1270 (11.1)	<0.001
Hypertriglyceridemia	624 (9.1)	976 (8.5)	0.241
Smoking			<0.001
Never smoked	6142 (89.6)	10,368 (91.2)	
Ex-smoker	418 (6.0)	351 (3.0)	
Current smoker	293 (4.2)	648 (5.7)	
BMI (kg/m^2^) ≥ 23	3635 (53.0)	5693 (50.0)	<0.001
Waist circumference (cm)	79.9 ± 10.3	79.7 ± 9.6	0.180
Aerobic activity rate	2702 (39.4)	4968 (43.7)	<0.001
Fasting glucose (mg/dL)	98.8 ± 22.0	98.3 ± 19.6	0.074
Triglycerides (mg/dL)	113.8 ± 77.2	112.3 ± 71.2	0.196
HDL-C (mg/dL)	54.0 ± 12.8	56.0 ± 13.3	<0.001
LDL-C (mg/dL)	115.8 ± 34.9	116.6 ± 35.4	0.436
AST (IU/L)	21.3 ± 11.6	22.1 ± 10.7	<0.001
ALT (IU/L)	18.0 ± 14.6	18.8 ± 14.0	<0.001
FIB-4 ≥ 1.45	1663 (24.2)	3013 (26.5)	<0.001
APRI ≥ 0.30	933 (13.6)	1878 (16.5)	<0.001

Continuous variables are expressed as means and standard deviations. Categorical variables are expressed as numbers and percentages. Abbreviations: BMI, body mass index; HDL, high-density lipoprotein cholesterol; LDL, low-density lipoprotein cholesterol; ALT, alanine aminotransferase; AST, aspartate aminotransferase; FIB-4, fibrosis index based on four factors; and APRI, AST-to-platelet ratio index.

## Data Availability

The original data presented in the study are openly available at https://knhanes.kdca.go.kr/knhanes/main.do (accessed on 19 September 2024).

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
