# Peer review of "Dietary Nutritional Supplements Are Associated with the Deterioration of Hepatic Fibrosis in Women and Individuals without Underlying Disease"

_ijerph, 2024, doi:10.3390/ijerph21101298_

Round 1
Reviewer 1 Report
Comments and Suggestions for Authors
The manuscript entitled “Dietary Nutritional Supplements are Associated with the Deterioration of Hepatic Fibrosis in Women and Individuals without Underlying Disease” is the study investigating the association between dietary supplement intake and hepatic fibrosis in South Korea using retrospective data from the national survey.
The way in which this topic, i.e. the possible negative effects of dietary supplements, is investigated is very interesting, but right at the beginning the question arises as to which and how many dietary supplements the participants took. In this general way, no general conclusions can be drawn about dietary supplements, especially given the wide range of supplements on the Eastern markets, with the emphasis on herbal supplements. Therefore, the work should be refined in particular to emphasise the dietary supplements taken by the respondents, i.e. whether they can be observed and compared as such.
Did you also analyse the normality of the data distribution in the statistics on data processing? Why were tests for comparing multiple groups of the ANOVA type not used where one would expect it given the data?
Figures 2 - 5 do not show us anything other than what you have provided in the text to explain. Maybe you could leave out image 2, but the others are really redundant. It would be better to present a comparison between the so-called control group and the group that took the supplements Also, it would be better if you showed and discuss the results of the supplements themselves (considering the type - vitamins, minerals, herbs, Herbalife, ...) and their influence.
Regarding the discussion chapter, it would be important to see what supplements are involved, what the literature says and what supplements were used by the respondents in your study. It would therefore be necessary to refer to the relevant literature that establishes a specific link between the consumption of dietary supplements and liver disease
You mention this as a limitation, but mostly the consumers of dietary supplements are conscious women, with a higher level of education, higher purchasing power, i.e. those who are somewhat older.
Comments on the Quality of English LanguageEnglish language is fine, minor text editing required
Reviewer 2 Report
Comments and Suggestions for Authors
Dear authors,
The manuscript submitted to me for evaluation raises a very important issue of the impact of supplements on the human body. The manuscript is written in a very interesting and clear way, but it lacks some information.
First of all:
- it is worth supplementing Table 1 with the significance of differences between genders and separating the control group from the study group.
- in Table 2, add columns with the division into women and men.
Supplementing this information will allow for a more complete analysis, because in the current form the informations in Tables 1 and 2 are given chaotically.
- Most of the discussion is poorly written - this is a description of the results and should be moved to another chapter. Please supplement the discussion with
references to studies by other authors.
- In addition, please supplement the list of literature with the latest studies - there is not enough of it, especially since this is a fairly common problem on the Internet.
After making corrections, the manuscript should be published.
Reviewer 3 Report
Comments and Suggestions for Authors
In the manuscript submitted to me for review entitled "Dietary Nutritional Supplements are Associated with the Deterioration of Hepatic Fibrosis in Females and in Subjects without Underlying Disease“ the authors Sangheun Lee, Minsu Cha and Kijun Han present a study investigating the relationship between hepatic fibrosis and the consumption of dietary supplements.
The conducted study could attract the attention of other researchers and contribute to the development of a new policy of stricter control and correct use of nutritional supplements.
The study included data from 30,639 patients who consented in advance to participate in the study, and all study protocols were approved by the National Center for Health Statistics Ethics Review Board.
To support their research, the authors used 19 references that present information from studies published mostly in the last two decades. Nearly 1/2 of the total references are from the last 5 years, indicating that interest in the causative agents of various liver diseases has increased over the years and the present manuscript would be of interest to IJERPH readers. I did not notice any redundant self-citations, all the references used are appropriate and necessary for the preparation of the manuscript.
My remarks and recommendations to the authors are:
1. On line 60, the statement is stated:
"various products manufactured by Herbalife (Los Angeles, CA, USA)"
In my opinion, the name of the company should be removed, as it may lead to damage to the reputation of the manufacturer concerned. I don't think the authors should commit themselves and IGERPH to presenting this information. The reference to Stckel et al. is quite sufficient if readers are interested in the information.
2. In the introduction, the authors report data on the increase in the use of dietary supplements in recent years. Do the authors have data on dietary supplement use in Korea?
3. Do the authors have data on the relationship between dietary supplements and the development of hepatic fibrosis in Korea so far, or is this the first such study?
4. It is good if the Introduction is expanded a little and more references are included. It would be helpful if you could describe hepatic fibrosis in a little more detail and include some data from Korea related to the issue at hand (for example, number of diagnosed patients with hepatic fibrosis over a period of time, types of dietary supplements most commonly used in Korea, or whatever information can be found).
5. In several places in the manuscript, the authors offer recommendations for controlling the use of dietary supplements. What are their next plans and strategies to continue working on the issue?
Round 2
Reviewer 2 Report
Comments and Suggestions for Authors
Thank you for the corrections. The manuscript is suitable for publication.